# Health Claims for Sports Drinks—Analytical Assessment according to European Food Safety Authority’s Scientific Opinion

**DOI:** 10.3390/nu16131980

**Published:** 2024-06-21

**Authors:** María Dolores Rodríguez-Hernández, Ángel Gil-Izquierdo, Carlos Javier García, José Antonio Gabaldón, Federico Ferreres, Daniel Giménez-Monzó, José Miguel Martínez-Sanz

**Affiliations:** 1Research Group on Quality, Safety, and Bioactivity of Plant Foods, Department of Food Science and Technology, CEBAS-CSIC, 30100 Murcia, Spain; mdrh@um.es (M.D.R.-H.); cjgarcia@cebas.csic.es (C.J.G.); 2Molecular Recognition and Encapsulation Group (REM), Health Sciences Department, Universidad Católica de Murcia, Campus Los Jerónimos 135, 30107 Murcia, Spain; jagabaldon@ucam.edu (J.A.G.); fferreres@ucam.edu (F.F.); 3Department of Community Nursing, Preventive Medicine and Public Health and History of Science Health, University of Alicante, 03690 Alicante, Spain; dgimenez@ua.es; 4Department of Nursing, Faculty of Health Sciences, University of Alicante, 03690 Alicante, Spain; josemiguel.ms@ua.es

**Keywords:** nutrition, sport, sports drinks, EFSA, health claims, fraud

## Abstract

In Europe, sports food supplements (SSFs) are subject to specific laws and regulations. Up to 70% of athletes are highly influenced by the information on the label or the advertisement of the SSF, which often does not correspond to the scientific evidence, such as health claims. The aim is to analyze such claims relating to sports drinks (SDs) in commercial messages. To this end, an observational and cross-sectional study was conducted based on the analysis of the content and degree of adequacy of the health claims indicated on the labelling or technical data sheet of the SDs with those established by the European legislation in force according to the European Food Safety Authority (EFSA). The SSFs were searched for via Amazon and Google Shopping. A total of 114 health claims were evaluated. No claim fully conformed with the recommendations. A total of 14 claims (n = 13 products) almost conformed to the recommendations; they were “Maintain endurance level in exercises requiring prolonged endurance”, “Improve water absorption during physical exercise”, and “Improved physical performance during high intensity, high duration physical exercise in trained adults”, representing 12.3% of the total (n = 114). The vast majority of the claims identified indicated an unproven cause–effect and should be modified or eliminated, which amounts to food fraud towards the consumer.

## 1. Introduction

### 1.1. Sports Food Supplements

Ergogenic aids have been defined as substances or methods used to improve endurance, overall fitness level, and sports performance. There are five categories of ergogenic aids: nutritional, pharmacological, physiological, mechanical, and psychological [1,2]. In the field of nutrition, foods and food components that can enhance an individual’s ability to exercise have also been described as nutritional ergogenic aids [3]. In the sports context, they are known as sports supplements (SSs) or sports food supplements (SSFs) and have been used for various purposes: in particular, to increase energy, maintain strength, health, and the immune system, improve performance, and prevent nutritional deficiencies [4,5,6,7].

The mechanisms by which SSFs can influence the body are extensively explained in the scientific literature and an example of this can be found on the website of the Australian Institute of Sport [8].

According to the International Olympic Committee (IOC), they are defined as “a food, compound, nutrient or non-food component that is purposefully ingested as part of the normal diet with the aim of achieving a specific health or performance effect”. These substances are consumed due to the competitive nature of sports, where athletes seek to achieve certain goals relating to SSFs [9]. They can be consumed by up to 90% of athletes, depending on the sport, and are a common practice in most athletes [6,10,11,12]. Specifically, the most commonly used supplements, in order of prevalence, are: protein powder or protein bars (66%), sports replenishment drinks (49%), creatine (38%), recovery drinks (35%), and vitamin C (25%) [6]. SDs are consumed by almost half of all athletes and play an important role in hydration [13]. According to the Australian Institute of Sport, the above SSFs are in evidence group A (compatible with use in specific situations in sport, using evidence-based protocols), with the exception of vitamin C and amino acids which are in group B (they require further research and should be used under specific research or clinical monitoring protocols) [8].

### 1.2. Sports Nutrition Supplements, Scientific Evidence and Legislation

The consumption of SSFs by athletes is conditioned by specific laws, regulations [14], instructions, and research by institutions such as the Australian Institute of Sport (AIS). Such legislation should provide advice or recommendations on the use and consumption, dosage, safety, precautions, and warnings of these substances [14]. It should also provide information on their market access and availability, as well as their efficacy with respect to enhancing sports performance. These are general principles of public health action to ensure that the population can achieve or maintain the highest level of health [15]. However, some popular products (glutamine, L-carnitine) are marketed as SSFs despite the lack of objective (scientific) evidence to support claims of an ergogenic effect [16].

The standards and regulations on SSFs vary between countries and product types. In the European Union and its member states, several provisions on sports nutrition can be found. They all include labelling (health or performance claims), safety and marketing aspects, and the content of vitamins, minerals, and other substances [17,18]. Currently, the legislation related to the regulation and application of ergogenic nutritional aids or sports nutrition products can be found in the following documents: Regulation (EU) No 1169/2011, Regulation (EC) No 353/2008, Regulation (EC) No 1924/2006, Regulation (EC) No 1925/2006, Directive 2002/46/EC, and Regulation (EU) No 609/2013. Despite the existence of this legislative framework, the regulations lack a regulatory sector governing the use and application of SSFs by consumers [14].

Within the SSFs, and included in the aforementioned regulations and directives, are the SDs. Scientific organizations and public institutions such as the European Food Safety Authority (EFSA) have previously studied the characteristics of the different substances added or isolated in supplements, as well as the safety of their consumption, and among these substances are the so-called SDs [19].

SDs are often chosen by athletes for hydration purposes, as they provide carbohydrates (CHs) and sodium (Na) to maintain the body’s homeostasis and prevent exercise-associated hyponatremia, which occurs in up to 10.3% of participants [20,21]. Dehydration is associated with a decrease in the ability of runners to maintain an even pace during a competitive situation, as shown in several studies, such as that of L. Stearns [22]. In addition, another important reason for its consumption is thermoregulation [23,24,25,26], the increase in internal body heat production, when developing in hot environmental conditions, must be controlled by different strategies (among them, acclimatization and hydration) to maintain the body homeostasis, which will allow the athlete to be more successful [22,23,27,28,29].

### 1.3. Sports Drinks

Commercial solutions designed specifically for exercisers and used during physical exercise are generally referred to as “drinks for sports” or “sports drinks”. They have a specific composition in order to achieve the rapid absorption of water and electrolytes, and to replenish CHs lost during activity [24]. Sports drinks can be used at different times during training or competition; their main purpose is to replenish fluids and electrolytes, so they are useful during training [30]. EFSA has issued opinions in favor of health claims that relate to SDs and that can be used for consumer information [19].

These claims are central to the marketing of the product, both in its labelling and in the advertising of the product for athletes [25]. These electrolyte and carbohydrate replenishment drinks, used in the appropriate quantities and proportions as indicated above, will serve to replenish fluid, electrolyte, and substrate losses, improving sports performance and reducing the risk of heat-related illnesses [24], although research in this field has detected cases in which the claims are not entirely in line with the described effects of certain food supplements on the health of the consumers [26].

Regarding the same SDs, there are also studies showing that nutrition labelling differs from the actual amount of ingredients. Fraud due to labelling errors, due to omission of substances present in the product or due to errors in the analysis or declaration of quantities, is high [31]. Market regulation is complicated by the growing popularity of internet sales. The need for quality control of products to ensure that they contain the listed ingredients in the stated quantity and to ensure the absence of potentially harmful substances is recognized. For most SSFs, the evidence is weak or even completely lacking. Difficulties arise when new evidence emerges to support novel supplements. Athletes rarely wait until there is convincing evidence of efficacy or safety, but caution is necessary to minimize risk [6].

To the best of our knowledge and as far as SDs are concerned, it has not yet been assessed whether the health claims on the label or advertising of SDs are in line with current scientific evidence.

Therefore, the objectives of this study are to describe the health claims of the SDs in commercial messages of a sample of SFFS, verify the adequacy of these health claims in accordance with current European legislation, and the accuracy of the information provided to the consumer.

## 2. Materials and Methods

### 2.1. Type of Study

An observational, cross-sectional study was carried out based on the analysis of the content and degree of adequacy of the health claims indicated on the labelling or technical data sheet of the replacement drinks with those established by current legislation and the evidence described to date. In addition, the study design, as well as the development of the manuscript, followed the STROBE statement [32].

### 2.2. Study Population Selection Strategy

The search for the sample products was conducted in December 2022 via the web shopping platforms, Amazon and Google Shopping. These websites were selected because they are the main online shopping websites. To carry out the search process, the terms “sports drinks”, “Carbohydrate-electrolyte drinks”, and “sports drinks” were entered on both portals. From this initial search, supplements that were only SDs were selected. Subsequently, we redirected to each of the websites of the selected supplement brands to obtain the health claims for each of them (see Appendix A). The process of obtaining each component of the sample was different depending on the portal visited.

### 2.3. Inclusion Criteria

In this study, SSFs defined as SDs and offered for sale in Europe were part of the selected sample. SSFs that were not defined as SDs, or that appeared several times within the search, on the same website, or both, were excluded.

### 2.4. Data Extraction

After carrying out the search to select the study sample, a descriptive analysis of the product characteristics of each of the selected SD products referenced in the labelling was carried out.

The variables studied for each product in the sample were as follows:Product name: The name of each of the supplements belonging to the study sample.Sports brand: The brand of each of the supplements in the sample. SSFs belonging to the sample were defined.Health claims for the SDs: Those on the labelling or website of each of the supplements in the requested sample.Dosage: The manufacturer’s recommended amount of consumption for each supplement.Nutritional information: The value of the nutrients, carbohydrates, protein, fat, and sodium in each of the sample supplements.

### 2.5. Data Analysis

After carrying out the data extraction, an analysis was performed, classifying the health claims indicated on the product or website, according to the approved health claims.

The following variables were obtained in this analysis:Approved Health Claims: EFSA-approved health claims for replacement drinks.Total supplements in which this declaration was made (no. and %): The total number and percentage of the supplements belonging to the sample in which this claim was made.Meet Conditions of Use: Conditions based on Regulation (EC) No 432/2012 [32] and Regulation (EC) No 686/2021 [33].No. and % of supplements in which these conditions of use were given for this claim: The number and percentage of supplements belonging to the sample in which the conditions of use were given for each claim.Health claims stated on the product or website: The health claims stated on each of the supplements in the sample.No. and % of supplements where the declaration and the statement appeared: number and percentage of supplements belonging to the sample in which the label or website declaration or statement appeared.Degree of adequacy statement yes/no: Whether the health claims for each of the supplements in the sample were adapted to the health claims defined by EFSA.Reason: According to EFSA scientific opinion, the reason for compliance or non-compliance and the proposal for modification of the supplements belonging to the sample to achieve a better adaptation to the health claims approved. Numbered from 1 to 7, from not compliant to compliant, respectively.

### 2.6. Compliance with Legislation and Scientific Evidence

Following the content analysis of the labelling of the products in the selected sample, a comparison of the different health claims made for the SDs in the labelling was carried out in order to determine their compliance with the scientific evidence established by EFSA (Table 1).

## 3. Results

The search yielded 160 results, of which 66 SSFs belonging to different commercial brands met the inclusion criteria set out in the methodology. Thirty-eight were rejected because they were duplicates on the same or both websites, and fifty-six were rejected because they were not replacement beverages (Figure 1). For the 66 selected SSFs, the health claims present and their dosage were specified.

### 3.1. Indicated Conditions of Use of the Product

As can be seen in Table 2, with respect to the conditions of use of each product, and the statement “Maintain endurance level in exercises requiring prolonged endurance”, 61.1% (n = 11) meet all three conditions of use. Regarding the statement “Improve water absorption during physical exercise”, 65.5% (n = 38) meet all three conditions of use. With regard to the statement “Improvement of physical performance during high intensity and high duration physical exercise in trained adults”, no product fulfils exactly the two conditions of use. A total of 27% (n = 10) meet both criteria almost completely, and 45.9% (n = 17) partially meet both criteria.

### 3.2. Health Claims and Compliance with Current Legislation and Scientific Evidence

Table 3 sets out a percentage distribution of each health claim found in the sample of SSFs and the type of health claim indicated by the manufacturer on each SSFs.

The most frequently described health claim in the SSFs is related to “Hydration (promotes, facilitates, maintain, improves, helps, optimal, recommended need hydration)” which is found in 36.4% of the supplements in the sample, followed by “Improving performance” and “Improves water absorption during exercise” with 25.8% and 19.7%, respectively, followed by the statements “Endurance: maintain level, improve endurance, increase endurance, satisfy endurance, maximise endurance” and “Maintaining performance” with 13.6%, followed by “Improved water absorption” and “Intensive and prolonged physical exercise recommended” with 9.1%. “Ensures water balance during training” is at 7.5%.

The following statements are at 6.1%, “Recommended for use during intensive and prolonged physical exercise” and “Water supply”. “Maintaining endurance performance during longer training” and “Increases water absorption during physical exercise” are at 4.5%. And “Suitable for intensive exercise” and “Muscle recovery after high intensity physical exercise” are at 3%.

In contrast, there are statements that only appear on a single product in the sports supplement sample, such as “Maintaining endurance performance during prolonged endurance exercise”, “Maintaining performance during prolonged endurance sessions”, “Maintaining sporting performance during prolonged endurance exercise”, “Improved fluid intake”, “Water absorbed quickly”, “Perfect for quenching thirst”, “Optimal muscle performance”, and “It can make a difference to your athletic performance”. These assume in each case 1.5% of the statements of the total sample.

### 3.3. Degree of Compliance and Proposals for Modification

The reasons for the adequacy or inadequacy of the claims of the analyzed SSF products, as well as the possible modifications to be made, are set out in Table 4 on the basis of what has been established by the scientific reference institutions (Table 1).

According to Table 4 (summary of distribution of conditions of use (dosage, etc.) of products according to health claims and adequacy of health claims), no SSF products with the claims “Maintain endurance level in exercises requiring prolonged endurance”, “Improve water absorption during physical exercise”, and “Improve physical performance during high intensity and duration physical exercise in trained adults” fully met the adequacy reasons of the claims.

SSF products bearing the claims “Maintain endurance level in exercises requiring prolonged endurance”, “Improve water absorption during physical exercise”, and “Improve physical performance during high intensity and duration physical exercise in trained adults” almost fulfilled the adequacy reasons of the claims representing 5.3%, 5.2%, and 27%, respectively, in each type of claim.

## 4. Discussion

In the present study, different health claims of SSFs included in the labelling or website of a sample of products were analyzed, as well as the conditions of use indicated for the achievement of these effects. In the results, no products were found to meet all the required conditions of use and the degree of compliance with the health claims. Products were found that met all the required conditions of use but did not fully comply with the degree of compliance with the health claim (score number 7) where the health claims identified were “Maintain endurance level in exercises requiring prolonged endurance”, corresponding to 5.2% (n = 3) of the products and the claim “Improve water absorption during physical exercise”, corresponding to 5.2% (n = 3) of the products. And for the claim “Improve physical performance during high intensity and duration physical exercise in trained adults”, full compliance was not met in 27% (n = 10) of products. All products analyzed have to modify the conditions of use or the adequacy of the required health claim. A total of 10.8% (n = 4) of products were found that despite naming benefits of the claim “Improved physical performance during high intensity and duration physical exercise in trained adults” did not comply with any of the conditions of use.

### 4.1. Health Claims and Proposed Doses

Sports drinks are designed to provide a balanced amount of carbohydrate, electrolytes, and fluids to allow the athlete to rehydrate and recharge simultaneously during and after exercise [8]. Carbohydrate consumed during exercise can support or enhance performance via two different mechanisms: the provision of fuel for muscle and a mouthfeel benefit to the brain and central nervous system [8]. The type and amount of carbohydrates contained in sports drinks varies by manufacturer, taking into account factors such as taste, osmolarity (concentration of individual particles), intestinal absorption, and intestinal tolerance [8].

The increase in sales of SSFs globally can be attributed, in part, to aggressive marketing by manufacturers, rather than nutritional supplements becoming more effective, and the accuracy of labelling is often not questioned [35]. One study found that 52.8% of websites do not provide scientific references for the products being marketed [36]. Another study found that just over 50% of respondents attached importance to the quality of the nutritional supplement information found on the package label. Nutrition and health claims and advertising, such as ergogenic claims, should be required by authorities and consumers to be supported by scientific evidence [37,38].

In this study, the different health claims present in the SSF sample have been compiled, with 33 different health claims proposed by the manufacturers, of which only 12 claims are fully or partially in line with what is established by the institutions. These claims are approved at a European level by both the EFSA and the European Commission (EC) [36], in addition to establishing specific legislation to allow the regulation of sports nutrition products and their advertising through consensus documents, and there is a register of health claims permitted in the marketing of these specific products [39].

On the other hand, it has been noted that in the claim “Improvement of physical performance during high intensity and duration physical exercise in trained adults”, no product reported in detail that only trained adults exercising (at least 65% of VO_2_max) and of long duration (at least 60 min) obtain the benefit, as established by scientific reference institutions [34].

Regarding the conditions of use on sodium (must provide between 20 mmol/L (460 mg/L) and 50 mmol/L (1150 mg/L)) in products with the claims “Maintain endurance level in exercises requiring prolonged endurance” and “Improve water absorption during physical exercise”, it has been observed that this required dose is not met in all SSFs, only 57.6% (39 products out of 59) of the total, despite being an essential mineral for proper fluid replacement [40]. Sodium is the only electrolyte that has been shown to be essential to help maintain a situation of eunatremia and, therefore, to maintain hydration in sport as a contributor to muscle function. Its recommended use is in the form of drinks and as an ingredient in supplements [30].

According to the EFSA [19], drinks specially designed for athletes must have a specific composition to achieve the rapid absorption of water and electrolytes, and prevent fatigue, and one of their main objectives is to replenish electrolytes, especially sodium, which also improves glucose absorption in the intestine. The recommended intake is between 460 mg/L and 1150 mg/L. According to the Australian Institute of Sport [8], the electrolyte content of sports drinks, especially sodium, helps to preserve thirst. Sodium concentrations of ~10 to 25 mmol/L improve palatability and voluntary intake of fluids consumed during exercise, although higher sodium/electrolyte concentrations may increase fluid retention.

In addition, sodium intake, in both high and low doses, is associated with health and performance problems in athletes. There are theories that an electrolyte imbalance, specifically sodium, contributes to the development of muscle cramps and hyponatremia [41]. Despite the importance of this mineral, in our study, 30.5% (18 products out of 59) of the SSFs analyzed did not meet the minimum recommended dose of sodium and 3.4% (2 products out of 59) exceeded the maximum recommended dose. The European Food Safety Authority (EFSA) has established a dietary reference value for this mineral at 2 g/day for the general population, pregnant women, and lactating women. In the case of children, the value has been extrapolated from that of adults and adjusted according to the different energy demands [42]. For athletes, a daily intake value has not been established, which may be similar to that of the general population. However, recommendations for the intake of this mineral during training and/or competition have been established. These recommendations indicate that in events lasting more than two hours, especially in hot/humid conditions, an intake of 300–600 mg Na/hour should be taken through food and/or SSs [8,9,21].

Regarding the conditions of use on osmolarity (should provide between 200 and 330 mOsm/kg of water) in products with the claims “Maintain endurance level in exercises requiring prolonged endurance” and “Improve water absorption during physical exercise”, it has been observed that 93.2% (55 products out of 59) of the products do not provide this information. Osmolarity is not a mandatory criterion to be reported, and according to Regulation 1169/2011 on the provision of food information to consumers [43], companies may choose not to include it in their products. However, it should be recommended that the company includes it because it is a criterion to be met in health claims [33] and in the criteria for a sports or isotonic drink [8,30].

Osmolarity, as well as sodium concentration and the form and amount of carbohydrates, are necessary to maximize gastric emptying and intestinal absorption of a sports drink during exercise, and have been the subject of debate and experimental research for more than two decades [44,45,46].

### 4.2. Fraud in Advertising and Direct-to-Consumer Information

As can be seen, product advertising does not always correctly refer to the effects of a particular food. Food fraud can be found in various forms in advertising, consumer information, and marketing of sports supplements, also observed in the study on the analytical assessment of health claims related to caffeine dosage in the labelling of sports supplements [47] and the study on health claims relating to creatine monohydrate [48].

The choice of SSFs must be made on the basis of criteria of safety, legality, and efficacy. There are several risks that some of these products can trigger; by way of example, the presence of a legal substance in doses higher than its recommendation may cause unwanted side effects; on the other hand, its presence in quantities lower than necessary (effective threshold) may not have the effect expected or announced by the product, leading to consumer fraud [31]. There are cases in which different SSFs have been shown to contain substances not reported on the label, higher or lower doses than those reported, or other types of contamination capable of damaging the athlete’s health, sporting performance, or sporting life, causing them to test positive for unintentional doping [12,49].

The product label represents a way of influencing the consumer’s purchasing decision, being significantly influenced by the information presented on the label [38]. It is important to consider that many athletes decide to use SSFs as part of their nutritional strategy, most of the time advised by people not qualified in sports nutrition such as coaches, teammates, or family and friends [50]. In addition, athletes who consume SSFs are not aware of any platform to check the safety/quality of SSFs [11].

### 4.3. Dealing with Advertising Fraud

The EFSA is involved in food safety in the context of public health at a European level, from the point of view of advertising and marketing of food products. In addition, regulation by means of legislative documents serves as a legally binding tool against advertising and food fraud [14]. There is also cross-cutting legislation, generally European in scope, directly applicable or incorporated into each legal system, which regulates advertising lawfulness and good commercial practices in general or for certain media and communication channels.

All this legislation places particular emphasis on the need for communications to be truthful, not to mislead users, and, in health matters, to be subject to scientific evidence and claims authorized by health authorities. In the case of food and advertising related to SSFs, the criteria established by WADA for the presence of prohibited substances in food should also be taken into account [49].

In Spain, there are associations for regulating food advertising, in addition to the work of the competent health and consumer authorities and the work of the courts, the advertising industry, and the media. These associations are the Association for the Self-regulation of Commercial Communication (Autocontrol), an independent self-regulatory body for the advertising industry in Spain, set up in 1995 and the Asociación de Usuarios de la Comunicación (AUC), which defends the interests of citizens in their relationship with the different media and communication systems and with the new information technologies, allowing any user to report any advertising content they consider illegal [51,52].

### 4.4. Cases of Advertising Fraud

As mentioned above, despite proposed legislation and pressure from governments and various organizations, such as the European Food Safety Authority (EFSA), WADA, and the International Olympic Committee, fraud still exists in the marketing of SSFs at three levels: health claims in product advertising, mislabeling, and adulteration/presence of substances banned by WADA [14,53].

Several studies have demonstrated fraud in supplement labelling, especially related to protein, creatine, and weight loss which may be due to unintentional or voluntary adulteration, or other contamination [54].

In the case of the study on health claims for caffeine labelling, the results show that only 2.78% of the claims fully comply with the cause–effect relationship established by the scientific reference documents, with the vast majority indicating an unproven cause–effect, which constitutes food fraud against the consumer [47]. Or, in the study on health claims for creatine monohydrate in commercial communications, where only 25% of the health claims met the criteria set by scientific reference documents. Most of the claims should be modified or removed as they could be considered fraudulent and/or misleading to the consumer [48].

Like other studies, where fraud due to labelling errors is high, either by omission of substances present in the product or by errors in analyzing or declaring the quantities of some of its ingredients. Inaccurate labelling and the omission of substances compromises consumer health and sports performance [31].

### 4.5. Limitations of the Study

One of the difficulties of this study was the variability in the search results, as well as the existence of products that did not provide the information required for this study. This work also highlights the multitude of health claims made by manufacturers or advertisers, in some cases presenting very confusing information. On the other hand, this study focuses on the European context (EFSA scientific opinion). Future research could extend this work. In particular, the European context on this issue could be analyzed and compared.

## 5. Conclusions

No claim fully conforms (scoring out of 8) with the recommendations. A total of 14 claims almost conformed (scoring out of 7) to the recommendations; they were “Maintain endurance level in exercises requiring prolonged endurance”, “Improve water absorption during physical exercise”, and “Improved physical performance during high intensity, high duration physical exercise in trained adults”, representing 12.3% of the total (n = 114). This should be compared with 100 claims that do not comply (scoring from 1 to 6) with the health claims established by the EFSA, representing 87.7% of the total (n = 114). The most used statements on the market are: “hydration (facilitates, facilitates, improves, helps, optimal, recommended, need hydration), found in 31.8% of the supplements in the sample, followed by “improves performance” and “improves water absorption during physical exercise” with 25.8% and 19.7%, respectively. Only the statement “improves water absorption during physical exercise” fits into the health statements. Health claims on SDs should be strongly aligned with the criteria established by legislation, consensus documents, and scientific evidence. Food fraud is found in various forms in food advertising, marketing, and commercialization, strongly affecting consumers. The achievement of quality food advertising should be the work of both advertisers and consumers, who should defend their rights under current regulations.

## Figures and Tables

**Figure 1 nutrients-16-01980-f001:**
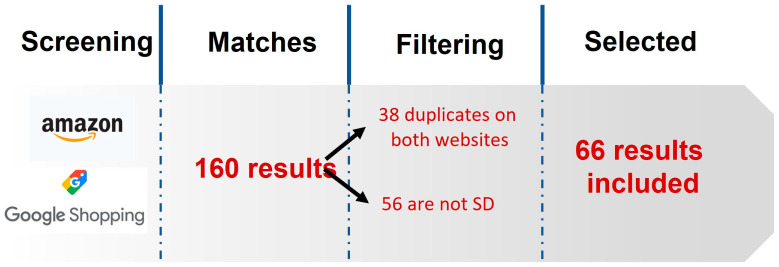
Flow chart showing how the study sample was obtained. SD: sports supplements.

**Table 1 nutrients-16-01980-t001:** Effects and applications of replacement drinks as established by EFSA’s scientific opinion.

	Types of Beverage Replenishment	Conditions of Use	Ergogenic Effects
EFSA Regulation (EC) No 432/2012 [33].	Carbohydrate electrolyte solutions	Between 80 and 350 kcal/L of carbohydrate. At least 75% of energy must be derived from carbohydrates. Sodium between 20 mmol/L (460 mg/L) and 50 mmol/L (1150 mg/L). Osmolarity between 200 and 330 mOsm/kg water.	Maintain the level of endurance in exercises that require prolonged endurance. Improve water absorption during physical exercise.
EFSA Regulation (EC) No 686/2021 [34].	Carbohydrate solutions	e. Between 30 and 90 g carbohydrate/hour, where the carbohydrate in question is glucose, sucrose, fructose, or maltodextrin, under these conditions: - Fructose (from fructose or sucrose) should not account for more than one third of the total carbohydrate, and - Glucose (from glucose, sucrose or maltodextrin) should not exceed 60 g/h. f. The consumer should be informed that only trained adults performing high-intensity (at least 65% of VO2max) and long-duration (at least 60 min) physical exercise and of long duration (at least 60 min) will obtain the beneficial effect.	Improved physical performance during high intensity and long duration physical exercise in trained adults.

**Table 2 nutrients-16-01980-t002:** Distribution of conditions of use of products according to health claims and their appropriateness.

Approved Health Claim	Total Supplements in Which This Declaration Is Made		Meet Conditions of Use *	No. and % of Supplements in Which These Conditions of Use Are Given for This Claim	
	No.	%			
Maintaining the level of resistance in exercises requiring prolonged endurance	19	28.8%	Meet 4 criteria: a., b., c., and d.	1	5.3%
			Meet 3 criteria: a., b., and c.	10	52.6%
			Meet 3 criteria: a., b., and d.	1	5.3%
			Meet 2 criteria: a. and b.	7	36.8%
Improve water absorption during physical exercise	58	87.9%	Meet 4 criteria: a., b., c., and d.	3	5.2%
			Meet 3 criteria: a., b., and c.	35	60.3%
			Meet 3 criteria: a., b. and d.	1	1.7%
			Meet 2 criteria: a. and b.	19	32.8%
Improved physical performance during high intensity, high duration physical exercise in trained adults	37	56.1%	Meet 2 criteria: a. and b.	0	0%
			Fulfils 2 criteria: a. and b. almost completely	10	27%
			Meet 2 criteria: a. and b. partially	17	45.9%
			Meet 1 criterion: a.	6	16.2%
			They do not meet any of the criteria	4	10.8%

* Conditions based on Regulation (EC) No 432/2012 [33] and Regulation (EC) No 686/2021 [34] listed in Table 1. Note: a sports nutrition supplement may have several nutrition claims.

**Table 3 nutrients-16-01980-t003:** Distribution of conditions of use of products according to health claims and their adequacy.

Approved Health Claim	Health Claims Stated on the Product or Website	No. of Supplements Where the Declaration Appears	% of Supplements Where the Declaration Appears	* Degree of Adequacy Statement Yes/No
Maintaining the level of resistance in exercises requiring prolonged endurance	Endurance: maintain level, improve endurance, increase endurance, satisfy endurance, and maximize endurance	9	13.6%	No
	Recommended for use during intensive and prolonged physical exercise	4	6.1%	No
	Maintaining endurance performance during longer training	3	4.5%	No
	Maintaining endurance performance during prolonged endurance exercise	1	1.5%	Yes
	Maintaining performance during prolonged endurance sessions	1	1.5%	No
	Maintaining sporting performance during prolonged endurance exercise	1	1.5%	No
Improve water absorption during physical exercise	Hydration: favors, facilitates, maintains, improves, helps, optimal, recommended, needed	24	36.4%	No
	Improved water absorption during exercise	13	19.7%	Yes
	Improved water absorption	6	9.1%	No
	Ensures water balance during training	5	7.5%	No
	Water supply	4	6.1%	No
	Increases water absorption during physical exercise	3	4.5%	No
	Improved fluid intake	1	1.5%	No
	Water absorbed quickly	1	1.5%	No
	Perfect for quenching thirst	1	1.5%	No
Improved physical performance during high intensity, high duration physical exercise in trained adults	Improving performance	16	25.8%	No
	Maintaining performance	9	13.6%	No
	Intensive and prolonged physical exercise recommended	6	9.1%	No
	Suitable for intensive exercise	2	3%	No
	Muscle recovery after high intensity physical exercise	2	3%	No
	Optimal muscle performance	1	1.5%	No
	It can make a difference to your athletic performance	1	1.5%	No

* Adequacy rating if the health claims for the replacement drinks indicated in the selected sample of supplements are adapted to the health claims defined by EFSA.

**Table 4 nutrients-16-01980-t004:** Summary distribution of conditions of use (dosage, etc.) of products according to health claims and adequacy of health claims.

Approved Health Claim	Total Supplements in Which This Declaration Is Made		Meet Conditions of Use	No. of Supplements in Which These Conditions of Use Are Given for This Claim	% of Supplements in Which These Conditions of Use Are Given for This Claim	Degree Of Adequacy Statement Yes/No	Reason *
	No.	% Total					
Maintaining the level of resistance in exercises requiring prolonged endurance	19	28.8%	Meet 4 criteria: a., b., c., and d.	1	5.3%	no	7
			Meet 3 criteria: a., b., and c.	1	5.3%	yes	6
				4	21.1%	no	5
				5	26.3%	no	4
			Meet 3 criteria: a., b., and d.	1	5.3%	no	4
			They meet 2 criteria: a. and b.	2	10.5%	no	1
				5	26.3%	no	2
Improve water absorption during physical exercise	58	87.9%	Meet 4 criteria: a., b., c., and d.	3	5.2%	no	7
			Meet 3 criteria: a., b., and c.	11	18.9%	si	6
				8	13.8%	no	5
				16	27.6%	no	4
			Meet 3 criteria: a., b., and d	1	1.7%	no	4
			Meet 2 criteria: a. and b.	2	3.4%	si	3
				2	3.4%	no	2
				15	25.9%	no	1
Improved physical performance during high intensity, high duration physical exercise in trained adults	37	56.1%	Meet 2 criteria: e. and f.	0	0	-	-
			Fulfils 2 criteria: e. and f. almost completely	10	27%	no	7
			Meet 2 criteria: e. and f. partially	17	45.9%	no	5
			Meet 1 criterion: e.	6	16.2%	no	5
			Do not meet any of the criteria	4	10.8%	no	1

Note: one supplement may have several nutrition claims. Source: authors’ own elaboration based on search data. * Reasons according to EFSA scientific opinion: Reason 1: not in accordance with the approved claims for carbohydrate electrolyte solutions. Proposed modification: delete product claim. Reason 2: not in line with the approved claims, but meets half of the recommended doses of the product. Proposed modification: amend the product claim and dosage regimen. Reason 3: Conforms to the approved claims, but does not meet half of the recommended dosages of the product. Proposed modification: modify dosage schedule of the product. Reason 4: Conforms to the approved claims and almost the recommended doses of the product, but does not specify type of exercise performed. Proposed modification: modify the declaration specifying the type of exercise in which the declared effects are shown and specify all the dosage guidelines of the product. Reason 5: conforms to the approved declarations, and almost the recommended doses of the product, but some do not indicate the effects that the supplement produces and quantities, or some detail of the declaration. Proposed modification: modify the claim by specifying who obtains the beneficial effect and amounts or the exact text, specifying all the dosage guidelines of the product. Reason 6: conforms to the approved claims, but lacks a specific recommended dosage of the product. Proposed modification: specify the dosing pattern of the product. Reason 7: Conforms to the approved claim and all recommended doses of the product but lacks modification of some detail of the claim or inclusion of amounts of the beneficial effect. Proposed modification: modify the claim by specifying the amounts of beneficial effect or the exact wording. Reason 8: In line with all of the above. Proposal for amendment: no change, no deletion of the claim.

## Data Availability

The data presented in this study are available in the tables and Appendix A of this article. The data presented in this study are available on request from the corresponding authors.

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
