# Peer review of "Health Claims for Sports Drinks—Analytical Assessment according to European Food Safety Authority’s Scientific Opinion"

_nutrients, 2024, doi:10.3390/nu16131980_

Round 1
Reviewer 1 Report
Comments and Suggestions for Authors
Dear authors,
I have carefully studied the manuscript entitled “Health claims for sports drinks. Analytical assessment according to EFSA's scientific opinion” by María Dolores Rodríguez-Hernández et al. The topic is very interesting but some adjustments need to be made:
1. A diagram showing the mechanisms by which SSFs can influence the function of the body would be useful.
2. line 59-60 Australian Institute of Sport is write two times.
3. can you give us more information about the use of sodium in the european legislation. What is the recommended quantity of Na?
4. can you tell us about the other electrolytes that we can found in the drinks? They help the body or can make a lot of damage? What is the recommendation of the UE forum?
5. line 343-345 please detail the adverse reaction of SSF
6. when it is necessary to use the SSF? before the exercise, in time or after? it will be interesting to know that. what is the perfect quantity for our body? it is different for men or women? but the age is important?
7. line 380 - you don't have tab
8.line 402 - you talk about the US legislation, but in the article you don't present the aspects
9. you can improve the conclusion with more informations about the electrolytes
Author Response
I have carefully studied the manuscript entitled “Health claims for sports drinks. Analytical assessment according to EFSA's scientific opinion” by María Dolores Rodríguez-Hernández et al. The topic is very interesting but some adjustments need to be made:
- A diagram showing the mechanisms by which SSFs can influence the function of the body would be useful.
Response of the authors: We are grateful for the reviewer's suggestion, but the authors believe that as this is not a review work, the diagram is not necessary and we have based ourselves on the EFSA criteria. However, we have indicated in the introduction that the mechanisms by which supplements can influence the body are amply explained in the scientific literature and an example of this in a visual way can be found on the website of the Australian Institute of Sport.
- line 59-60 Australian Institute of Sport is write two times.
Response of the authors: Following the reviewer's suggestions, the sentence about repeated Australian institute of Sport has been deleted to the Introduction.
- can you give us more information about the use of sodium in the european legislation. What is the recommended quantity of Na?
Response of the authors: Following the reviewer's suggestions, a paragraph has been added to the discussion section on European-level partner recommendations (lines 349-358).
- can you tell us about the other electrolytes that we can found in the drinks? They help the body or can make a lot of damage? What is the recommendation of the UE forum?
Response of the authors: We are grateful for the reviewer's suggestion, but the authors have not expanded the information on the other electrolytes found in beverages because we have based ourselves on the criteria mentioned by EFSA for claims. We agree with what he requests but such an analysis could be very lengthy and would not give us time to present a proper paper. The types of minerals that can be included in beverages are varied, with sodium always being common, and there are other types of minerals that are added to beverages, which are not the objective of this work.
- line 343-345 please detail the adverse reaction of SSF
Response of the authors: Following the reviewer's suggestions, the adverse reaction of SSF has been added to the Discussion.
- when it is necessary to use the SSF? before the exercise, in time or after? it will be interesting to know that. what is the perfect quantity for our body? it is different for men or women? but the age is important?
Response of the authors: We are grateful for the reviewer's suggestion, but the authors have focused not on the timing but on the use of the drinks, and that they are mainly focused on the during. We have added in Introduction that the main purpose of the drinks is fluid and electrolyte replenishment so they are useful during training.
- line 380 - you don't have tab
Response of the authors: Following the reviewer's suggestions, the tab has been added to the text.
8.line 402 - you talk about the US legislation, but in the article you don't present the aspects
Response of the authors: Following the reviewer's suggestions, the US legislation has been deleted to the text.
- you can improve the conclusion with more informations about the electrolytes
Response of the authors: Following the reviewer's suggestions, but the authors believe that the focus of this work is on EFSA health claims.

Reviewer 2 Report
Comments and Suggestions for Authors
The manuscript analyzes the health claims on sports drinks according to the EFSA scientific opinion. The topic is highly relevant, as the authors have pointed out, the popularity of sports nutrition products is increasing. For athletes, food labels are one source of information, so it is important that they receive reliable information on which they can base their consumer decisions.
The manuscript analytically examines the health claims, and the structure of the manuscript as a whole and its parts are logical and well-organized. The study design meets the research objectives, and the methodology describes the selection of sports drinks and data extraction. The discussion is appropriate for the results, and the authors make important, forward-looking statements.
However, the methodology section lacks a description of the analysis method, which makes it difficult to interpret the results. Therefore, I recommend considering the following minor modifications:
It would be worthwhile to add a paragraph to the methodology under the title 'data analysis,' where the authors describe what the column names in the tables of the results section actually mean. In Tables 2, 3, and 4, there are asterisked sections noted after the table to aid understanding. Without reading these, the tables are very difficult to interpret, so I suggest including these parts in the methodology section as well, so readers can be informed in advance.
In Table 1, it would also be useful to standardize the notation in the 'condition of use' column, as currently, the parts are marked in various ways: a-, (a), b). This is important because the other tables build on this table. For example, in Table 4, the 'Comply with Conditions of Use' column refers to criterion C, but in Table 1, there is no criterion C, only a and b."
Additionally, it would be worthwhile to mention that the health claims stated on the product or website were classified according to the approved health claims. At least, this is what appears in Table 3.
Author Response
The manuscript analyzes the health claims on sports drinks according to the EFSA scientific opinion. The topic is highly relevant, as the authors have pointed out, the popularity of sports nutrition products is increasing. For athletes, food labels are one source of information, so it is important that they receive reliable information on which they can base their consumer decisions.
The manuscript analytically examines the health claims, and the structure of the manuscript as a whole and its parts are logical and well-organized. The study design meets the research objectives, and the methodology describes the selection of sports drinks and data extraction. The discussion is appropriate for the results, and the authors make important, forward-looking statements.
However, the methodology section lacks a description of the analysis method, which makes it difficult to interpret the results. Therefore, I recommend considering the following minor modifications:
It would be worthwhile to add a paragraph to the methodology under the title 'data analysis,' where the authors describe what the column names in the tables of the results section actually mean. In Tables 2, 3, and 4, there are asterisked sections noted after the table to aid understanding. Without reading these, the tables are very difficult to interpret, so I suggest including these parts in the methodology section as well, so readers can be informed in advance.
Response of the authors: Following the reviewer's suggestions, the paragraph under the title 'data analysis,' has been added to the Methodology, where the columns names in the tables 2, 3 and 4 of the results have been described.
In Table 1, it would also be useful to standardize the notation in the 'condition of use' column, as currently, the parts are marked in various ways: a-, (a), b). This is important because the other tables build on this table. For example, in Table 4, the 'Comply with Conditions of Use' column refers to criterion C, but in Table 1, there is no criterion C, only a and b."
Response of the authors: Following the reviewer's suggestions, in Table 1, the notation in the 'condition of use' column has been standardized.
Additionally, it would be worthwhile to mention that the health claims stated on the product or website were classified according to the approved health claims. At least, this is what appears in Table 3.
Response of the authors: Following the reviewer's suggestions, Following the reviewer´s suggestions, the sentence "the health claims stated on the product or website were classified according to the approved health claims" has been mentioned.

Round 2
Reviewer 1 Report
Comments and Suggestions for Authors
Dear authors, I read the modified article and I think that wright now can be published. The subject is very interesting and useful for many people.